# A Novel Colorimetric Sensor Array Coupled Multivariate Calibration Analysis for Predicting Freshness in Chicken Meat: A Comparison of Linear and Nonlinear Regression Algorithms

**DOI:** 10.3390/foods12040720

**Published:** 2023-02-07

**Authors:** Wenhui Geng, Suleiman A. Haruna, Huanhuan Li, Hafizu Ibrahim Kademi, Quansheng Chen

**Affiliations:** 1School of Food and Biological Engineering, Jiangsu University, Zhenjiang 212013, China; 2Department of Food Science and Technology, Kano University of Science and Technology, Wudil, P.M.B 3244 Kano, Kano State, Nigeria; 3College of Food and Biological Engineering, Jimei University, Xiamen 361021, China

**Keywords:** colorimetric sensor array, chicken freshness, TVB-N, multivariable algorithm, linear and non-linear models

## Abstract

As a source of vital nutrients for the normal functioning of the body, chicken meat plays an important role in promoting good health. This study examines the occurrence of total volatile basic nitrogen (TVB-N) as an index for evaluating freshness, using novel colorimetric sensor arrays (CSA) in combination with linear and nonlinear regression models. Herein, the TVB-N was determined by steam distillation, and the CSA was fabricated via the use of nine chemically responsive dyes. The corresponding dyes utilized, and the emitted volatile organic compounds (VOCs) were found to be correlated. Afterwards, the regression algorithms were applied, assessed, and compared, with the result that a nonlinear model based on competitive adaptive reweighted sampling coupled with support vector machines (CARS-SVM) achieved the best results. Accordingly, the CARS-SVM model provided improved coefficient values (*Rc* = 0.98 and *Rp* = 0.92) based on the figures of merit used, as well as root mean square errors (RMSEC = 3.12 and RMSEP = 6.75) and a ratio of performance deviation (RPD) of 2.25. Thus, this study demonstrated that the CSA paired with a nonlinear algorithm (CARS-SVM) could be employed for fast, noninvasive, and sensitive detection of TVB-N concentration in chicken meat as a major indicator of freshness in meat.

## 1. Introduction

Food safety should be a primary focus for food experts as the world’s population expands and consumer demand for food rises. Chicken, with its high protein and low fat content, is a popular food in many countries owing primarily to its widespread availability [1]. However, one significant concern is the short shelf life of chicken flesh. This is because unprocessed chicken flesh is extremely ephemeral and may only be stored for three to five days at 4 °C in the refrigerator [2]. Additionally, the distribution channels, retail outlets, and end users, on the other hand, have a vested interest in products that can be stored for an extended time without compromising quality or safety [3,4]. Therefore, due to the fact that chicken meat consumption is growing annually, there is a global food safety issue with chicken meat that may have an impact on consumer health [5]. To address these obstacles, an effective technique for the fast and precise monitoring of freshness in chicken meat is necessary.

To date, chemical, physical, and microbiological tests, as well as sensory evaluation, are the most important means for the traditional assessment of freshness in chicken [6]. Several critical indicators have been applied for determining the freshness of meat and other aquatic species such as the thiobarbituric acid reactive substances (TBARS), the total viable count (TVC), and total volatile basic nitrogen (TVB-N). Most importantly, chemical and microbiological examinations, including total viable counts, pH analysis, and TVB-N assay, are often quite straightforward [2,7]. The downside of these approaches is that they are frequently damaging, tedious, and not suitable for use in conjunction with contemporary industrial processing and production techniques. As a result, the swift and noninvasive assessment of chicken freshness is becoming particularly crucial. Furthermore, sensory evaluation methodologies based on human sensations have widespread market appeal [3]. However, human, time, and geographical issues all have an impact on the precision and reliability of the outcomes. Consequently, sensory tests are subjective and are based on the degree of satisfaction of a particular community due to its customs and geographical location. Hence, the demand for quick and noninvasive chicken freshness monitoring is becoming deeply vital [8]. The odor of chicken meat is one of the most essential criteria to consider when assessing its freshness. In addition, its nutritional richness makes it an ideal colonization substrate for many disease and spoilage bacteria [8,9]. During microbiological deterioration, microbes and enzymes break down the proteins and lipids in chicken meat, progressively decreasing its freshness. A diverse number of VOCs are produced by enzymes and microbes during the breakdown of primary materials such as hydrogen sulfide, ethanal, and hydrogen nitride [10].

Several sophisticated analytical instruments, such as the electronic tongue (E-tongue), computer vision (CV), electronic nose (E-nose), and various forms of spectroscopic technology, have recently evolved for the noninvasive analysis of food quality and safety [3,11]. The E-nose, sometimes called the artificial olfactory approach, is made up of an array of numerous generic sensors that generate a unique fingerprint in response to an odor stimulus [12]. Additionally, the vast majority of commercially accessible artificial olfactory devices nowadays are based on metal oxide semiconductor (MOS) sensors. Conductometric behavior is typical of MOS sensors; this implies that the resistance of the sensor varies, either decreasing or increasing, when it is exposed to odor vapor molecules [13]. In this regard, the MOS sensor array is restricted by the fact that it is sensitive to changes in humidity.

Herewith, a novel artificial olfactory method based on CSA proves its overriding feature of eliminating humidity interference and emerges as a potent instrument for the evaluation and monitoring of food quality [14,15]. Printing chemical reaction dyes onto inert substrates is one of the core principles underlying this method. These dyes can include pH indicators, dyes comprising huge permanent dipoles, metal salts, dyes involving metal ions, and nanoparticles [16]. It differentiates by utilizing the color shift caused by the interaction between the VOCs and the array of chemically sensitive dyes upon ligand binding. As a method for food evaluations using multivariate calibration, we retrieved the values of RGB (red, green, and blue) color constituents from color change patterns for each dye. The CSA strategy proved effective in identifying intricate food matrices in a variety of foods, including fruits, vegetables, tea, bakery, and meat products [17,18].

Chemometrics techniques enable the investigation of complex signals. To analyze data, many researchers turn to linear regression algorithms, with partial least squares (PLS) being one of the most used [19,20]. This technique, however, retains several irrelevant and redundant variables, lowering model predictive strength and precision. In this scenario, the PLS-coupled spectral variable selection algorithms have the potential to produce better analytical results [21]. Due to the complexity of color change profiles, linear regression models may be unable to provide sufficient explanation. Further, nonlinear regression models have a greater capacity for self-learning than linear regression algorithms [22]. As a result, in this work, the nonlinear regression tool “support vector machine (SVM)” is used to address the regression issue as an expeditious learning neural method with minimal training error.

The primary goal of this work was to fabricate a visible CSA in combination with chemoselective responsive dyes paired with chemometric algorithms for real-time TVB-N measurement in chicken meat samples. The scope of the research covers the following areas: (1) a chemo-responsive dye was printed onto reverse-phase silica gel plates for preparation of the CSA sensors; (2) color change patterns of VOCs emitted from chicken meat samples were acquired using a CSA at various storage times (1, 3, 5, 7, 9, and 11 days); (3) by subtracting the images of the CSA sensor acquired before and after the sample was exposed, a color-distinctive fingerprint of the chicken meat samples was obtained; and (4) multivariate statistical methods were applied in order to qualitatively and quantitatively analyze the TVB-N through robust modeling using four different efficient variable selection algorithms based on linear and nonlinear regression models. Figure 1 is a diagrammatic depiction of the proposed approach.

## 2. Experimental Sections

### 2.1. Collection and Preparation of Chicken Samples

The collection and utilization of chicken meat samples were based on the Chinese National Standard (GB/T 32762-2016). Adult Luyuan (LY) chickens, one of China’s most popular chicken breeds and a native of Zhangjiagang city in Jiangsu Province of China, were used with the following specifications: Age = 10 weeks; antemortem live weight = 2550~2750 g (2.75 kg); postmortem time = 45 min. The chicken breast from the same batch was received fresh, sealed in clean polyvinyl chloride (PVC) bags, ice-packed, and delivered to the laboratory as soon as possible. The samples were sliced into sections (4 cm × 3 cm × 0.5 cm) and then measured into equal weights of 10 g (ca. ±0.1 g) on a sterile cutting board to enable daily sampling in subsequent tests and to reduce probable mistakes. Further, given the heterogeneous and complicated nature of chicken meat, all samples were placed, sealed, labeled, and packaged into separate clean plastic bags and kept under refrigeration at 4 °C. During the 11-day storage period, 14 samples were randomly taken out from the refrigerator to determine their TVB-N content every other day (i.e., 1st, 3rd, 5th, 7th, 9th day, and 11th day). Given the inevitable systematic errors in individual samples, 80 chicken breast samples were finally separated into calibration and prediction sets in a 3:2 ratio for the process of model building.

### 2.2. Reference Measurement of TVB-N Content

The TVB-N is a significant chemical and physical test used to determine the quality of meat. In this experiment, the TVB-N concentration was determined using steam distillation in accordance with the Chinese National Standard (GB/T 5009.228-2016) with minor adjustments [8]. To begin with, 10 g (±0.1 g) of chopped sample was placed in a beaker, mixed with 100 mL of distilled water, and vigorously shaken for 30 s using a high-speed disperser XHF-DY, Xinzhi, Inc., Ningbo (Zhejiang, China). We then filtered the mixture solution using filter paper after allowing it to stand for 30 min at room temperature. Then, 10 mL of filtrate and 5 mL of Magnesia suspension (10 g/L) were both distilled for 5 min using a Kjeldahl distillation apparatus, and distilled water amounting to 10 mL was used as a standard. A 50 mL aqueous solution of boric acid (40 g/L) and a solution of 0.1 g bromocresol green and methyl red in ethanol (95%, 100 mL) were both added to the distillate before it was collected in an Erlenmeyer titration flask. A final titration of 0.1 mol/L HCl was performed on the boric acid solution. Each sample was tested in triplicate. The TVB-N results were expressed in mg/100 g, and the concentration was calculated adhering to the following formula:(1)$$X\;=\frac{{(v}_{1}\;{\;-\;v}_{2})\;\times \;c\;\times \;14}{m\;\times \;5/100}\;\times \;100$$
where X is the sample’s TVB-N concentration represented in mg/100 g, *v*_1_ is the quantity of hydrochloric acid ingested by the titrated boric acid absorbing liquid (mL), *v*_2_ is the quantity of hydrochloric acid absorbed by the titrated blank absorbing liquid (mL), *c* represents the concentration of HCL (mol/L), and *m* denotes sample weight in grams. The values of the TVB-N content during refrigerated storage (1–11 days) were shown in Appendix A.

It is worth noting that the TVB-N measurements, as well as the picture collection before and after the silica gel plate reaction, were all performed on the same day with the same samples. This was specifically performed to ensure the correctness and consistency of the results.

### 2.3. Fabrication of the Colorimetric Sensor Array (CSA)

The chemically responsive dyes selected for this experiment were tested prior to the experiment to make sure the colorimetric sensor array could provide an optimum response. As shown in Appendix A, this study employed twelve chemoresponsive dyes (comprising three pH markers and nine metalloporphyrins). Analytes with Lewis acid-base capabilities can be recognized by porphyrins and their metallic complexes with a good deal of accuracy. The metalloporphyrins are almost perfect for the identification of metal-ligating vapors due to their high color intensity, significant spectrum shifts upon ligand binding, and free coordination points for axial ligation. The indicators of pH are dyes that change color when their environment changes from being acidic or basic according to the proton (Brønsted) acidity [3,15].

In order to achieve the successful fabrication of the odor imaging sensor array, it is imperative to select the most suitable plate. For this research, C2 reverse-phase silica gel plates from Merck KGaA in Darmstadt, Frankfurt, Germany, were used. The colorimetric sensors were created using the four procedures. To begin with, the pH indicators were dissolved in ethanol after a precise amount of 20 mg of each chemically responsive dye was diluted in a 10 mL chloroform solution. The mixture was then ultrasonically treated for 2 h at room temperature to produce a solution (2 mg/mL). As a result, each dye solution was then printed onto the plates using microcapillary pipettes of 0.1 µL capacity. Thereafter, the CSA, consisting of nine pigment solutions (3 × 3 dyes), was constructed. The fabricated arrays were kept in a nitrogen-flushed glove bag before being used in this experiment.

### 2.4. Collection of CSA Data

In this study, an HP Scanjet 4890 flatbed scanner was used to capture the images obtained by the CSA (Hewlett Packard Inc., Shanghai, China). A resolution of 600 dots per inch (dpi) was selected on the scanner to provide sharp and detailed images. The original image was taken before a chicken meat sample was scanned using a flatbed scanner, avoiding any potential exposure to the sensor array. After that, a sample of chicken was placed in front of a colorimetric array. Here, a 250 mL glass vessel was used to contain the sample so that it could come into contact with the sensor array. This allowed for the best possible reaction between the CSA and the head gas produced by the free volatile compounds in each meat sample. A ventilator provided the necessary exposure for the duration of this analysis, the room temperature remained constant at 25 °C, and the sample was kept at 4 °C prior to data collection. The following criteria were organized in a logical order: the size of the Petri dish, the duration of the headspace, and the volume of the sample. Thus, the CSA was removed from the glass vessel and rescanned after thorough equilibration to acquire the final image. As depicted in Figure 1, we subtracted the final image from the initial image to develop a vibrant difference image. This difference image was used to determine the color change delineation of volatile oxidative compounds (VOCs) in the sample [16,17]. It should be mentioned that exploratory tests were carried out in this work to estimate the equilibration period of the sensor response. Accordingly, preliminary investigations showed that after 5 min of equilibration, the desired reactions between the VOCs and the dyes were achieved. As a result, the response time was fixed at 5 min. Finally, non-uniformity was eliminated by averaging the centers of each dye point (each point is a circular region made up of 800 pixels).

### 2.5. Multivariable Calibration Analysis

#### 2.5.1. Theory of Partial Least Squares (PLS) Model

PLS is a straightforward classical linear chemometric technique that has found widespread use, notably in the evaluation and identification of analytes of interest [21,23]. The PLS relies heavily on the complete spectrum data set, which comprises both crucial and irrelevant data sets. Consequently, the inferior performance of the created PLS model is deemed a result of the utilization of redundant and unimportant variables.

#### 2.5.2. Theory of Support Vector Machine (SVM) Model

The support vector machine (SVM) is a nonlinear model that is often employed for functional regression analysis [24]. The SVM is a very effective machine-learning approach that was developed from the theory of statistical learning. There is a growing application of this technique in many fields, such as classification (SVC) and regression analysis (SVR), as it is able to decipher practical challenges including small samples, nonlinearity, and high dimensionality in a practical manner [24,25].

#### 2.5.3. Theory of Random Frog (RF) Model

The RF is one of the most structured and efficient variable filtering approaches, and it was first designed to identify genes and categorize diseases [10,21]. This variable screening method algorithm is based on the reversible jump Markov Chain Monte Carlo (RJMCMC) method. The RF variable selection model is mathematically clear and has been demonstrated to be computationally efficient as it utilizes an iterative process. Based on three stages, this model works as follows: (1) the variables in the subgroup *V*_0_ that include the *Q* variables are chosen at random; (2) iteratively, the subset’s identified and favored variables are enhanced by variables constructed from a Norm (Q, θQ). The Q and θQ represent the mean and standard deviation, respectively; and (3) after selecting the variables, a probability is calculated from which the variables’ suitability and significance can be determined.

#### 2.5.4. Theory of Uninformative Variable Elimination (UVE) Model

Centner et al. were the first to propose the UVE algorithm. It is a prominent method for selecting relevant variables by taking into account the robustness of the PLS regression model [26]. The elimination of the extraneous variables is supported by an increase in the variance of the dependent variable in conjunction with a decrease in that variable’s covariance (y). The UVE approach uses the matrix X (N ×) and the matrix R (N × L) of artificial random variables to assess the importance of each prediction variable (*k*). The stability value and the cutoff threshold are the two most critical aspects for the execution of this model. A Monte Carlo cross-calculation is used to calculate the frequency of the maintained variables in the former, whereas the artificial random variables with the highest possible absolute value of c are used in the latter. This cutoff is used to exclude variables that provide no useful information, such as those with absolute *c* values below the threshold. The PLS regression equation’s coefficient matrix is analyzed in UVE using a leave-one-out approach, and the resultant equation is as follows:(2)$$\hat{y}=Χ\beta +Ε$$
where $\hat{y}$ is the projected value vector (*n* × 1), which is composed of variables from n spectra, X is the matrix (*n* × *ρ*), *β* marks the vector regression coefficients (*ρ* × 1), and *E* is the model offset vector (*n* × 1).

#### 2.5.5. Theory of Competitive Adaptive Reweighted Sampling (CARS) Model

CARS is a frequently employed variable selection method that is beneficial for locating and utilizing factors that are pertinent to the analytes being examined [16]. In particular, variables with strong definite coefficients in PLS are called “important spectral variables.” Thus, the importance of the variables is determined by the absolute values of the regression coefficients generated by PLS [20]. For this reason, CARS relies on the sequential, competitive, and iterative selection of crucial spectral characteristics, including: (1) based on the actual coefficients of the relevant variables, a subset of N Monte Carlo sampling runs was selected for each cycle of CARS; (2) using the exponentially declining function (EDF) and adaptive reweighted sampling (ARS), the variables with small true regression coefficient values were excluded; and (3) the best subset for variable selection was established by estimating the root mean square error of calibration (RMSEC) for each potential subgroup.

### 2.6. Statistical Data Analysis

As part of this study, the major usage of Matlab R2014b was involved for the analysis of spectrum and reference data, as well as model applications, which were all run on Microsoft Windows 10 (Mathworks Inc., Natick, MA, USA). The variable selection approaches such as RF, UVE, and CARS were applied based on linear (PLS) and nonlinear (SVM) algorithms.

## 3. Results and Discussion

### 3.1. Reference Measurement Results

In order to evaluate the freshness of chicken meat samples, the actual content of TVB-N was evaluated. Table 1 shows the descriptive statistics of these data. The results show that there was no discernible variation in the standard deviation of the samples from the calibration set and those from the prediction set. In addition, neither the range nor the mean of the samples was significantly different, indicating that both calibration and prediction sets were effectively distributed. This effective distribution of samples across the two data sets (calibration and prediction) was deemed sufficient and verified the model’s application for the efficient prediction of TVB-N contents in chicken meat samples.

### 3.2. Colorimetric Sensor Array Characteristics Variables (CSA)

The microbial deterioration of chicken flesh is distinguished by the production of nitrogenous compounds, otherwise referred to as VOCs. In this study, the concentration of TVB-N was found to be closely related to the volatile component amines emitted by chicken meat samples during refrigeration. The CSA plays an important role in the detection of the fundamental odor changes that take place in spoiled chicken meat. These changes were primarily caused by the action of microbes and the decomposition of specific intrinsically active compounds such as proteins, carbohydrates, and fat in the chicken [2]. The two primary reasons why the chosen chemical dyes respond to most VOCs by presenting definite colorific fingerprints are simplified molecular structure alteration and open coordination sites for axial ligation [3,10]. The color-sensitive image points collected in this study contain 10 variables altogether: red, green, and blue color (RGB), hue, saturation, and value (HSV), LAB values, and Euclidean distance. Further, the color-specific fingerprints can be recognized by the sensor by displaying a peculiar response. The CSA difference images were separated by subtracting the initial image from the final image following exposure to the VOCs of the chicken meat samples, resulting in a dramatic color shift for each individual sample. As clearly illustrated in Figure 1, the difference image of a CSA was obtained by digitally evaluating each pixel before and after the exposure. A unique fingerprint is created by the color of each image (∆R, ∆G, ∆B, ∆H, ∆S, ∆V, ∆L, ∆A, ∆B, and Euclidean Distance).

### 3.3. Sample Classification and Evaluation of the Model Performance

For the successful application of the prediction models, the samples (80 in total) were separated into calibration and prediction sets in a 3:2 ratio. During the analysis of the data sets, 48 samples were allocated to a calibration set for the purpose of constructing the model. The remaining 32 samples were assigned to a prediction set to test the robustness of the model that was constructed [10]. An SPXY technique, which is based on the distances between the X and Y variables, was utilized, and this resulted in the successful categorization of data into calibration and prediction sets [27].

Several statistical parameters were used and evaluated in this study to assess the effectiveness of the built models. Thus, the correlation coefficients of calibration and prediction (*Rc* and *Rp*), the root mean square errors of calibration and prediction (RMSEC and RMSEP), and the ratio of the performance deviation (RPD) of the created models were employed and compared [21,22]. The lower values of RMSEC and RMSEP and the higher values of *Rc* and *Rp* are therefore preferred when assessing the performance of the model. Additionally, the RPD value is typically utilized to evaluate the reliability of the developed models. As a result, RPD values of ≥1.50 indicate that the model is practicable, whereas values of ≥2.00 suggest that the model is stable [14]. These evaluation parameters were computed using the formula below:(3)$$Rc=\sqrt{1-\frac{\sum_{i=1}^{n}\left(y_{cal}-y_{act}\right)}{\sum_{i=1}^{n}\left(y_{cal}-y_{mean}\right)}}$$
(4)$$RMSEC=\sqrt{\frac{\sum_{i=1}^{n}{\left(y_{cal}-y_{act}\right)}^{2}}{n}}$$
(5)$$\;Rp=\sqrt{1-\frac{\sum_{i=1}^{n}\left(y_{pre}-y_{act}\right)}{\sum_{i=1}^{n}\left(y_{pre}-y_{mean}\right)}}$$
(6)$$RMSEP=\sqrt{\frac{\sum_{i=1}^{n}{\left(y_{pre}-y_{act}\right)}^{2}}{n}}$$
(7)$$RPD=\frac{SD}{RMSEP}\;$$
where, *n* is the number of samples employed for model development; *y_cal_* is the predicted value in calibration set; *y_act_* is the actual value measured by reference method; *y_mean_* represents the average value; *y_pre_* is the predicted value in prediction set, and *SD* is the standard deviation of measured values in prediction set.

### 3.4. Different Variable Selection Algorithms Based on Linear and Nonlinear Regression Models

#### 3.4.1. Results Variable Selection Algorithms Based on PLS Linear Regression Model

The PLS was utilized in the form of a traditional linear regression model, and it was based on a total of ninety variables from the fabricated CSA sensor. Appendix A and Table 2 show the outcome of the PLS model’s performance in predicting TVB-N in a chicken meat sample. Appendix A illustrates that a minimum of four nLVs were chosen to demonstrate comparably significant linear fits between CSA data and the respective reference data for TVB-N prediction. It is, however, evident from the scatter plot performance presented in Appendix A that the results achieved by this model are accompanied by high prediction error values (RMSEP mg/100 g) as well as low prediction coefficient values (*Rp*). It appears that the PLS model’s performance is influenced by the use of redundant variables deemed irrelevant for the analyte under study [16,23].

The RF-PLS algorithm was used as an efficient variable selection approach for predicting TVB-N in chicken flesh samples. Notably, prior to its operation, the RF-PLS basic configurations were N = 10,000 cycles, a baseline quantity of variables featured in the subset Q = 2, and a moderating value of variability = 0.2, as presented in Figure 2A [21]. Meanwhile, default settings were kept for the rest of the RF-PLS setup parameters. The best result for TVB-N using this model was seen when nLVs = 5 (Figure 3A and Table 2), with *Rp* = 0.81 and RMSEP = 9.65 mg/100 g, respectively (Figure 3B and Table 2). Additionally, the UVE-PLS variable selection technique was used to identify pertinent informative factors from the CSA data in order to build a reliable and accurate model for quantifying TVB-N in chicken meat samples. In the present study, UVE-PLS was employed utilizing a number of different selection criteria, as shown in Figure 2B. The *t*-value of the stability index is shown in blue (vertical line), and two dotted blue (horizontal line) depict the upper and lower bounds of the real and hypothetical variables, respectively. Furthermore, the real and generated variables are separated by a tiny blue line by yellow and red wavy lines on the left and right sides, respectively. The variable was determined to be quite steady when a coefficient of 0.99 was employed in the selection procedure. As demonstrated in Figure 3D and Table 2, the performance of this model provided an ideal value of *Rp* = 0.77 and RMSEP = 10.52 mg/100 g. This model was further developed using two latent variables, as can be seen in Figure 3C. Additionally, for the purpose of creating a robust quantitative model for TVB-N concentration in chicken flesh samples, the CARS-PLS variable selection method was employed to extract critical factors from the CSA data. The following CARS configurations were used for maximum efficiency: the Monte Carlo number of sample sessions was 25, the iteration count was 5, and the frequency of the coefficients of variance was 5 [16]. As illustrated in Figure 2C (intermediate plot), proper variables were proposed and developed in two stages to ensure precision in the CARS-PLS model: in the initial cycle, the number of investigated variables was rapidly reduced to display a rapid sampling of variables. The second stage involves loosening selection criteria so that the number of variables sampled steadily grows up to 50 (Figure 2C, last plot). Accordingly, as is evident from examining Figure 2C (initial plot), the RMSEC values of the selected variable segments dropped drastically and then continuously increased up to the 17th sampling. The CARS-PLS model outperformed the prior model (UVE-PLS) with an nLVs of 4, as seen in Figure 3E. Furthermore, as displayed in Figure 3F and Table 2, the correlation coefficient of the prediction and error values is *Rp* = 0.81 and RMSEP = 9.71 mg/100 g.

#### 3.4.2. Results Variable Selection Algorithms Based on SVM Nonlinear Regression Model

The SVM was used to establish a relationship between the CSA data and the reference chemical data for actual TVB-N content prediction in chicken meat samples prior to applying variable selection models based on the SVM nonlinear model. The SVM model uses RBF as the kernel function, and the predictive performance of the model will be affected by the penalty parameter (c) and the RBF kernel function parameter (g). A clearly depicted example of the optimization of the RMSEC model parameters (g and c) using the grid optimization algorithm and the five-fold cross-validation method is shown in Appendix A. The RMSEC value is the least when g = 0.25 and c = 2, as is clearly shown in the figure above. Accordingly, based on the CSA spectral features, this model is the optimum SVM prediction model and produced *Rp* and RMSEP (mg/100 g) values of 0.83 and 9.26, respectively, as indicated in Appendix A and Table 2. Subsequently, the SVM models served as the basis for the application of variable selection algorithms such as RF, UVE, and CARS. It should be noted that before implementing these variable selection approaches, the SVM models were also optimized for parameters g and c to ensure the lowest RMSEC for effective prediction performance [25]. As seen from Figure 4A, the lowest RMSEC for the RF-SVM model was achieved when g = 0.25 and c = 4, whereas the lowest RMSEC for the UVE-SVM model was obtained when g = 0.088 and c = 4 (Figure 4B). On the other hand, as seen in Figure 4C, the best RMSEC score for CARS-SVM was achieved with g = 0.5 and c = 2. As result, as shown by scatter plot performance in Figure 4D for RF-SVM, the best results were shown for variable selection fused with SVM, with *Rp* = 0.86 and RMSEP = 9.17 mg/100 g. Similarly, for UVE-SVM, an improved outcome was attained with *Rp* = 0.89 and RMSEP = 7.90 mg/100 g (Figure 4E and Table 2). In addition to this, the CARS-SVM, which was chosen as the optimal model, generated a better outcome with *Rp* = 0.92 and RMSEP = 6.75 mg/100 g, as can be plainly shown in Figure 4F and Table 2, respectively.

### 3.5. Discussion and Comparison of the Built Models

In the present study, we explored a novel CSA combined with algorithms to examine TVB-N as an indicator of freshness in chicken meat samples. The application of these algorithms relied on variable selection (RF, UVE, and CARS) approaches, along with PLS and SVM as linear and nonlinear models, to develop a more robust predictive model. In the initial application, the PLS as a linear regression model was utilized to establish a correlation between CSA data and reference chemical data for TVB-N. As shown in Table 2, correlation was established, but with low *Rc* and *Rp* values as well as strong RMSEC and RMSEP scores as validated by the RPD value (>1.00). This unconvincing performance of the PLS might be explained by the fact that it utilized entire variables, which are considered to have contained some information that was extraneous to the target analyte. Hence, variable selection techniques were used for the screening and exploitation of relevant variables to predict TVB-N in chicken meat samples. The variable selection approaches were used, and the results improved, especially for RF-PLS and CARS-PLS, as seen by the RPD values of 1.61 and 1.58 for the former and latter, respectively (Table 2). This result may be related to the use of informative factors that were thought to have contained data pertinent to the prediction of TVB-N in chicken meat samples. It can be shown that RF-PLS, UVE-PLS, and CARS-PLS selected 30, 18, and 13 variables, respectively, representing 33.33%, 20%, and 14.44% of the total variables (Table 2). A further note should be made that, despite the fact that the linear regression models (PLS) produced significantly better results than the TVB-N prediction, the result generally has been regarded as poor, as it has recorded low *Rc* and *Rp* values, as well as high RMSEC and RMSEP values, as clearly illustrated in Figure 3B,D,F and Table 2.

To investigate whether there is a nonlinear relationship between the TVB-N content and the CSA database, the Runs Test tool was employed for the datasets. The details about the features of the Runs Test tool are shown in the Appendix A. In particular, |z| > 1.96 shows that the analyzed data are nonlinear when the significance level is 0.05. The testing results, calculated to be z = 2.88 for TVB-N (Appendix A), revealed the strong nonlinearity between the TVB-N content and the CSA database in chicken. Thus, a nonlinear regression model known as the support vector machine (SVM) was utilized on the obtained CSA data, which included a total of 90 different factors, in order to predict TVB-N. As shown in Table 2, the SVM outperformed the PLS in terms of *Rc* and *Rp* values, as well as prediction errors (RMSEC and RMSEP), as evidenced by an RPD value of 1.52. The variable selection strategies (RF, UVE, and CARS) based on the SVM nonlinear regression model outperformed those based on the linear PLS (Table 2). The *Rc* and *Rp* increased dramatically, whereas RMSEC and RMSEP remained exceedingly low. Given that these models were applied using the same number of variables as the linear, as is vividly seen in Table 2, their performance is appreciably improved. Notably, the CARS-SVM model generated the best results, with the highest *Rp* and the lowest RMSEP. Furthermore, the CARS-SVM achieved the highest RPD value of 2.25 (Figure 4F and Table 2), indicating that it is the most accurate and robust model for predicting TVB-N in chicken breast samples.

## 4. Conclusions

In conclusion, a novel method using CSA in conjunction with linear and nonlinear regression models has been developed for detecting the freshness of chicken meat samples. The TVB-N content was determined through steam distillation, and the CSA was effectively and efficiently fabricated by employing nine chemoselective dyes. We discovered a good association between the volatile organic compounds (VOCs) released by the chicken flesh during refrigeration and the corresponding dyes employed after successfully fabricating the CSA. Then, variable selection techniques were used, such as random frog (RF), uninformative variable elimination (UVE), and competitive adaptive reweighted sampling (CARS). These techniques were based on linear “partial least squares” (PLS) and nonlinear “support vector machines” (SVM) regression models. As a result, the CARS-SVM, as a nonlinear model, offered the best results for predicting the analyte under study, with a high prediction coefficient (*Rp*) of 0.92 and the lowest prediction error (RMSEP) of 6.75. This great performance could be attributed to the selection of the fewest factors necessary for the prediction of TVB-N in chicken flesh samples, as evidenced by the highest RPD value of 2.25 obtained. Given the tremendous nutritional relevance of chicken meat, the proposed approach might be used to monitor the freshness of chicken meat as part of the important difficulties afflicting the meat industry and consumers. In spite of this, further research should be conducted to determine whether or not the developed technique can be applied to other types or breeds of chicken meat, particularly as chicken meat is not homogeneous in nature.

## Data Availability

Data is contained within the article.

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
