# Peer review of "A Novel Colorimetric Sensor Array Coupled Multivariate Calibration Analysis for Predicting Freshness in Chicken Meat: A Comparison of Linear and Nonlinear Regression Algorithms"

_foods, 2023, doi:10.3390/foods12040720_

Round 1
Reviewer 1 Report
The manuscript is well written and the topic is quite interesting and useful. However, in my opinion there is a main lack. The meat of chicken is not homogenous and the usefulness of the system depends on the origin of the samples. It must be provided the origin of the chicken, feeding, breed, weight and all the variables which define the kind of chicken used. In just a kind of chicken was used it must be included in discussion and conclusions as a limitation of the system. -The shelf life of meat are measured using different variables, but the most popular is the lipid oxidation measured with the TBARS method. Why did you use the TVB-N instead the amount of malondialdehyde? -In my opinion, it would be very interesting to show values of TVB-N at each time. -L181 How did you subtract the images? Please, explain. -L247 The statistical analysis statement must be more explicative. The methods must be reproducible by other authors. Please, include the algorithms used and all the details needed to replicate your study. - L280 why did you divide by 3? difference of color with the cielab space is different - Please improve the resolution of Fig. 2 - Please, reorder the Sib figures of Fig. 3 according the PLS components . - When the RF-PLS algorithm was used, did you made a cross-calibration or used both calibration and validation sets? Cross-validations are very limited and are not adequate to draw to conclusions. - Please, explain the parameters g and c.Author Response
Dear Respected Editor,
On behalf of my co-authors, we thank you very much for giving us an opportunity to revise our manuscript. We appreciate you and the reviewers for the positive and constructive comments on our manuscript entitled “A novel colorimetric sensor array coupled multivariate calibration analysis for predicting freshness in chicken meat: A comparison of linear and nonlinear regression algorithms” (foods-2175637).
According to the comments raised by the reviewers, we gave the corresponding responses and made revisions that were underlined in red. Attached please find the revised version, which we would like to submit for your kind consideration. Looking forward to hearing from you!
Thank you again for your consideration!
Yours sincerely,
Dr. Quansheng Chen
29th January, 2023.
Response to Reviewer 1
Reviewer #1:
The manuscript is well written and the topic is quite interesting and useful. However, in my opinion there is a main lack.
Re: The authors are grateful for the comments you have provided regarding our manuscript. Your suggestions are valuable and can be used to revise and enhance our paper as well as to guide future research. Several changes have been made based on the comments you have provided. The revisions are marked in red in the manuscript. The main corrections in the paper and the responses are as follows:
Q1: The meat of chicken is not homogenous and the usefulness of the system depends on the origin of the samples. It must be provided the origin of the chicken, feeding, breed, weight and all the variables which define the kind of chicken used.
Re: Thank you for your valuable suggestion. According to your comments, we have provided the detailed information about the origin of the meat sample and the variables which define the kind of chicken used. The collection and utilization of chicken meat samples were based on the Chinese National Standard (GB/T 32762-2016). Adult Luyuan (LY) chickens, one of China's most popular chicken breeds and a native of Zhangjiagang city in Jiangsu Province of China, were used with the following specifications: Age = 10 weeks; antemortem live weight = 2550 ~ 2750 g (2.75 kg); postmortem time = 45 min.
Actually, in this study, the color sensitive sensor used is constructed based on the response between odor information generated by the meat sample and chemical dyes. Thus, the same size and specification of 14 samples (4 cm × 3 cm × 0.5 cm) were randomly taken out from the refrigerator to determine its TVB-N content every other day (i.e. 1st, 3rd, 5th, 7th, 9th day and 11th day) and react with colorimetric sensor. In addition, we also optimized the reaction time of color sensitive dye and meat odor information to ensure adequate reaction and avoid the other errors as much as possible. The results demonstrated that the CSA coupled with CARS-SVM could be employed for fast, noninvasive, and sensitive detection of TVB-N content in chicken meat.
Certainly, we think that your suggestion is very valuable and very helpful for our future research. In the future, we will also explore the feasibility of our colorimetric sensor array using different chicken breeds and different sampling sites and expand the number of samples. Hence, according to your suggestion, to make the reader better understand, we have revised the description of the origin of the samples we used in this study more accurately. Kindly find the updated section 2.1 (Collection and preparation of chicken samples) in the revised manuscript for these changes.
Q2: In just a kind of chicken was used it must be included in discussion and conclusions as a limitation of the system.
Re: Thank you for your valuable suggestion. According to your suggestion, the limitations of the present method with respect to the kind of chicken breed used have been emphasized in the conclusion. Kindly find the Lines 460-462 in the revised manuscript for your noting.
Q3: The shelf life of meat are measured using different variables, but the most popular is the lipid oxidation measured with the TBARS method. Why did you use the TVB-N instead the amount of malondialdehyde?
Re: Thank you for your valuable and beneficial comments and suggestions. Indeed, the TBARS method and total volatile basic nitrogen (TVB-N) are two critical indicators for determining the freshness of meat and other aquatic species. The difference is that TBARS focus on evaluating lipid oxidation degree of samples, while TVB-N is a key indicator to directly reflect sample odor information.
However, in this study, the colorimetric sensor used is constructed based on the response between odor information generated by the meat sample and chemical dyes. Chicken is rich in nutrition and contains a variety of enzymes. The growth and propagation of bacteria and the action of enzymes will cause the odor change of chicken. Therefore, the odor information change of chicken is one of the most important criteria to evaluate its freshness. While in chicken rotten odor, mainly ammonia and amine substances, and these alkaline nitrogen substances collectively known as TVB-N. Therefore, TVB-N is often used as the key index for the freshness evaluation of meat products in national standards. Therefore, this is the main reason why TVB-N was selected to carry out the research in this study. In addition, more importantly, the main principle of the color sensitive sensor constructed in this study to detect the freshness of chicken is based on the response between chicken odor information and dyes. Therefore, the TVB-N that the main nitrogen-containing odors of chicken is selected as the freshness evaluation index, which is directly related to the content of this study.
Certainly, we think that your suggestion is very valuable and very helpful for our future research. In the future, our group is engaged in evaluating the freshness of chicken meat as well as other meats and aquatic life, exploring different indicators such as TBARS, TVC, and TVB-N in conjunction with other approaches. Kindly find the Lines 53-56, 66-72, 104-106 in the revised Mns.
Q4: In my opinion, it would be very interesting to show values of TVB-N at each time.
Re: Thank you for your valuable suggestion. According to your suggestion, we have added the values of TVB-N at each time. Kindly find the Figure S1 and Table S1 in the revised Supporting information.
Q5: L181 How did you subtract the images? Please, explain.
Re: Thank you for your valuable suggestion. In this study, MATALAB R2014b was used to extract the red, green, and blue color (RGB), Hue, Saturation, and Value (HSV), and LAB values for each point of the array before and after the reaction. The difference images were obtained by subtracting the pre-reaction image from the post-reaction image, i.e., the value after exposure to the sample VOCs minus the pre-exposure value for the (∆R, ∆G, ∆B, ∆H, ∆S, ∆V, ∆L, ∆A, ∆B and Euclidean Distance). Kindly find the section 3.2 (colorimetric sensor array characteristics variables (CSA)) in the revised manuscript for your perusal.
Q6: L247 The statistical analysis statement must be more explicative. The methods must be reproducible by other authors. Please, include the algorithms used and all the details needed to replicate your study.
Re: Thank you for your valuable and productive comments and suggestion. According to your comments, the statistical analysis statements have been improved with the addition of algorithms used. In addition, the figures of merit used for the evaluation of the built models were also provided to improve clarity. Please refer to the section 2.6 (statistical data analysis) and section 3.3 (sample classification and evaluation of the model performance) in the revised manuscript for your noting.
Q7: L280 why did you divide by 3? difference of color with the cielab space is different
Re: Sorry for our mistake. We have deleted the original formula. Actually, the color sensitive image points collected in this study contain 10 variables altogether. The difference images were obtained by subtracting the pre-reaction image from the post-reaction image, i.e., the value after exposure to the sample VOCs minus the pre-exposure value for the (∆R, ∆G, ∆B, ∆H, ∆S, ∆V, ∆L, ∆A, ∆B and Euclidean Distance). Kindly find section 3.2 (colorimetric sensor array characteristics variables (CSA)) in the revised manuscript for your perusal.
Q8: Please improve the resolution of Fig. 2
Re: Thank you for your insightful observation. According to your observation, we have improved the resolution of Figure 2 to make it more legible. Please find the improved Figure 2 in the revised manuscript.
Q9: Please, reorder the Sib figures of Fig. 3 according the PLS components.
Re: Thank you for your valuable suggestion. As per your suggestion, the subfigures of Figure 3 have been rearranged according to the PLS components. Kindly find the corrected Figure 3 in the revised manuscript for your review.
Q10: When the RF-PLS algorithm was used, did you make a cross-calibration or used both calibration and validation sets? Cross-validations are very limited and are not adequate to draw to conclusions.
Re: Thank you for your and constructive and valuable comments. In this study, we used both calibration and validation set in the process of building RF-PLS model. Specifically, the whole set of spectral variables (90), which included both the calibration and validation sets, in order to obtain an accurate result. As shown in Table 1, 48 samples were allotted to the calibration set and 32 samples were assigned to the validation set, using an SPXY approach for sample division based on a 3:2 ratio. Several statistical parameters were used and evaluated in this study to assess the effectiveness of the built models. Thus, the correlation coefficients of calibration and prediction (Rc and Rp), root mean square errors of calibration and prediction (RMSEC and RMSEP) of the created models were employed and compared. To make reader better understand, as you suggested, we have supplemented more details about the sample division and several statistical parameters we used in this study. Please refer to section 3.3 (sample classification and evaluation of the model performance) in the revised manuscript for your noting.
Q11: Please, explain the parameters g and c.
Re: Thank you for your valuable and fruitful comments, all in line to make this manuscript a better one. In this study, the establishment of SVM model uses RBF as the kernel function, and the predictive performance of the model will be affected by the penalty parameter (c) and the RBF kernel function parameter (g). c and g were optimized by using grid optimization algorithm and five-fold cross validation method, and the optimal parameters c and g were obtained, so as to establish the optimal CARS-SVM model. To make reader better understand, as you suggested, we have supplemented more explanation about the parameters g and c. Kindly find the Lines 371-375 in the revised Mns.

Reviewer 2 Report
Manuscript ID: foods- 2175637
Title: A novel colorimetric sensor array coupled multivariate calibration analysis for predicting freshness in chicken meat: A comparison of linear and nonlinear regression algorithms
Comments:
The manuscript is very interesting and presents information on a novel colorimetric sensor array coupled multivariate calibration analysis for predicting freshness in chicken meat. However, the writing contains some writing and editing errors. There are inaccuracies in the introduction and in materials and methods. It is not clear if the model predicts TVB-N in chicken meat samples. Why don't you show the correlation of instrumental measurements of TVB-N content in chicken samples with those predicted with the non-linear model? It would be expected that during refrigerated storage the TVB-N values would increase. No readings were made after model development. What level of precision in terms of detection limit does the model have? The following comments should be taken into account to enrich the writing.
Línea 4. Remove “and b*”
Line 9. Correct font size and remove hyperlink.
Line 51. Descriptive sensory tests are accurate, reliable, and analytical, and can be correlated with objective measurements. Clarify if they are referring to tests with consumers (acceptance and preference), which are subjective and depend on the level of satisfaction of the population according to customs and geographical areas. The magnitude and intensity of the attributes related to chicken freshness could be perfectly evaluated with a descriptive sensory test.
Line 112. Please include more details about the origin of the samples (breed, age, antemortem live weight, post-mortem time, type of packaging during receipt, whether the samples were refrigerated or frozen, weight, if the breasts come from the same batch, etc.), in order to ensure homogeneity in the experimental units. How many chicken breasts were used?
Line 114. Is this sample size representative for the whole chicken breast? Were various areas of the chicken breast considered?
Line 118. Aerobic packing? Describe packaging characteristics (material caliber, material, measurements).
Line 119. Explain where the 80 chicken samples came from. If there are 14 samples x 6 days, there should be 84 samples in total and not 80. Please mention in detail.
Line 260. And the values of the TVB-N content during refrigerated storage (1-11 days)?
Line 263. Why was the total viable count not determined to relate to the volatile nitrogenous compounds produced during refrigerated storage?
Author Response
Dear Respected Editor,
On behalf of my co-authors, we thank you very much for giving us an opportunity to revise our manuscript. We appreciate you and the reviewers for the positive and constructive comments on our manuscript entitled “A novel colorimetric sensor array coupled multivariate calibration analysis for predicting freshness in chicken meat: A comparison of linear and nonlinear regression algorithms” (foods-2175637).
According to the comments raised by the reviewers, we gave the corresponding responses and made revisions that were underlined in red. Attached please find the revised version, which we would like to submit for your kind consideration. Looking forward to hearing from you!
Thank you again for your consideration!
Yours sincerely,
Dr. Quansheng Chen
29th January, 2023.
Response to Reviewer 2
Reviewer #2:
The manuscript is very interesting and presents information on a novel colorimetric sensor array coupled multivariate calibration analysis for predicting freshness in chicken meat.
Re: Thank you for your valuable comments regarding our manuscript. The comments help us both to improve the paper and to lead our research in the right direction. After carefully reviewing your comments, we have made corrections that we hope you will find acceptable. The revised portions of the manuscript are highlighted in red. The main corrections and our responses to your remarks are as follows:
Q1: However, the writing contains some writing and editing errors. There are inaccuracies in the introduction and in materials and methods.
Re: Thank you for your positive and valuable comments. According to your suggestion, the introduction and the materials and methods section has been thoroughly improved for better clarity. Kindly find the updated introduction and materials and methods in revised manuscript.
Q2: It is not clear if the model predicts TVB-N in chicken meat samples. Why don't you show the correlation of instrumental measurements of TVB-N content in chicken samples with those predicted with the non-linear model?
Re: Thank you for your positive comments. In this study, we used instrumental analysis to measure the TVB-N value of chicken meat, and used CARS-SVM to construct a prediction model for the TVB-N content of chicken meat during 11 storage days. The results shown that the establish CARS-SVM model provided satisfying coefficient values (Rc = 0.98 and Rp = 0.92) between instrumental measurements of TVB-N content in chicken samples with those predicted with non-linear CARS-SVM, as well as root mean square errors (RMSEC = 3.12 and RMSEP = 6.75) and a ratio of performance deviation (RPD) of 2.25. Thus, this study demonstrated that the CSA paired with a nonlinear algorithm (CARS-SVM) could be employed for fast, noninvasive, and sensitive detection of TVB-N concentration in chicken meat. According to your suggestion, to make reader better understand, we have supplemented more details about correlation of TVB-N content and those predicted values with the non-linear model in this study. Kindly find the Lines 32-34, 428-435 in the revised Mns.
Q3: It would be expected that during refrigerated storage the TVB-N values would increase. No readings were made after model development. What level of precision in terms of detection limit does the model have?
Re: Thank you for your positive and constructive comments. In this study, the established CSA paired with a nonlinear algorithm (CARS-SVM) were employed for sensitive detection of TVB-N in chicken meat and achieve a high detection accuracy. The CSA paired with a nonlinear algorithm (CARS-SVM) did produce the better results with the higher Rp of 0.92, lower RMSEP of 6.75 along with the increased model precision and performance (see Table 1 in the MNs) as well.
Specifically, as shown from Table 1, a nonlinear model based on CARS-SVM achieved the best results. The CARS-SVM model provided improved coefficient values (Rc = 0.98 and Rp = 0.92) based on the figures of merit used, as well as root mean square errors (RMSEC = 3.12 and RMSEP = 6.75) and a ratio of performance deviation (RPD) of 2.25. It shows that the model has good detection precision and feasibility. Besides, we also used the T test analysis on the measurements of TVB-N content value and predicted value by CARS-SVM of the sample, P=0.47 (P> 0.05), indicating no significant difference between the predicted value and the actual detected value. The aforementioned results demonstrated that the colorimetric sensor array coupled with support vector machines (CARS-SVM), have good detection performance and are detection precision for freshness in meat. According to your suggestion, we have supplemented more details about the model precision and performance. Please kindly find the Lines 32-34, 428-435 in the revised Mns.
Q4: Line 4. Remove “and b*”
Re: Thank you for your positive and insightful observation. The “and” which shouldn’t there has been deleted.
Q5: Line 9. Correct font size and remove hyperlink.
Re: Thank you for your positive comments. The font size has been corrected and the hyperlink has also been removed.
Q6: Line 51. Descriptive sensory tests are accurate, reliable, and analytical, and can be correlated with objective measurements. Clarify if they are referring to tests with consumers (acceptance and preference), which are subjective and depend on the level of satisfaction of the population according to customs and geographical areas. The magnitude and intensity of the attributes related to chicken freshness could be perfectly evaluated with a descriptive sensory test.
Re: It is very much appreciated that you offered positive and constructive comments, all in line with improving the manuscript. The sentences have been clarified based on your ideas by adding a clearer explanation for a better comprehension of the context. Kindly find line 63-65 in the revised manuscript for your review.
Q7: Line 112. Please include more details about the origin of the samples (breed, age, antemortem live weight, post-mortem time, type of packaging during receipt, whether the samples were refrigerated or frozen, weight, if the breasts come from the same batch, etc.), in order to ensure homogeneity in the experimental units. How many chicken breasts were used?
Re: Thank you for your positive and helpful feedback. According to your suggestion, we have added the corresponding description about the origin of the samples. “The collection and utilization of chicken meat samples were based on the Chinese National Standard (GB/T 32762-2016). Adult Luyuan (LY) chickens, one of China's most popular chicken breeds and a native of Zhangjiagang city in Jiangsu Province of China, were used with the following specifications: Age = 10 weeks; antemortem live weight = 2550 ~ 2750 g (2.75 kg); postmortem time = 45 min. The chicken breast from the same batch was received fresh, sealed in clean polyvinyl chloride (PVC) bags, ice-packed, and delivered to the laboratory as soon as possible. The samples were sliced into sections (4 cm × 3 cm × 0.5 cm) and then measured into equal weights of 10 g (ca. ± 0.1 g) on a sterile cutting board to enable daily sampling in subsequent tests and to reduce probable mistakes. Further, given the heterogeneous and complicated nature of chicken meat, all samples were placed, sealed, labeled, and packaged into separate clean plastic bags and kept under refrigerated at 4 °C. For following 11 days, 14 samples were randomly taken out from the refrigerator to determine its TVB-N content every other day (i.e. 1st, 3rd, 5th, 7th, 9th day and 11th day).”
Certainly, we think that your suggestion is very valuable and very helpful for our future research. In the future we will consider conducting research based on different types and specific site of chicken and expand the number of samples for the chicken freshness prediction. According to your suggestion, we have incorporated the above details pertaining to the samples in order to provide a better understanding of the entire manuscript. Kindly find the updated collection and preparation of chicken samples (section 2.1) in the revised manuscript for scrutiny, which now read as.
References:
[1] Bao, W. B., Shu, J. T., Wu, X. S., Musa, H. H., Ji, C. L., & Chen, G. H. (2009). Genetic diversity and relationship between genetic distance and geographical distance in Chinese indigenous chicken breeds and red jungle fowl. Czech Journal of Animal Science, 54(2), 74-83.
[2] Zhang, X., Leung, F.C., Chan, D.K., Yang, G., Wu, C. (2002). Genetic diversity of Chinese native chicken breeds based on protein polymorphism, randomly amplified polymorphic DNA, and microsatellite polymorphism. Poultry Science, 81(10), 1463-1472.
Q8: Line 114. Is this sample size representative for the whole chicken breast? Were various areas of the chicken breast considered?
Re: Thank you for your positive and helpful comments. In this study, the color sensitive sensor used is constructed based on the response between odor information generated by the meat sample and chemical dyes. Thus, the same size and specification of 14 samples (4 cm × 3 cm × 0.5 cm) were randomly taken out from the refrigerator to determine its TVB-N content every other day (i.e. 1st, 3rd, 5th, 7th, 9th day and 11th day) and react with colorimetric sensor. Therefore, this study ensures the representativeness of sample selection and expands the scope of sample data set as much as possible by random sampling and take 14 parallel samples every day. In addition, we also optimized the reaction time of color sensitive dye and meat odor information to ensure adequate reaction and avoid the other errors as much as possible. Certainly, we think that your suggestion is very valuable and very helpful for our future research. In our future research, we will select the chicken samples from different regions, expand the sample numbers, comparing the difference of detection results in different regions of chicken breast, and build a quantitative prediction model of chicken freshness. Kindly find the updated section 2.1 (Collection and preparation of chicken samples) in the revised manuscript for these changes.
Q9: Line 118. Aerobic packing? Describe packaging characteristics (material caliber, material, measurements).
Re: Thank you for your positive and helpful comments. The packing we used is an aerobic packing since the packaging material does not entail any gas flushing into the container. This type of packaging was used in part due to the fact that samples are delivered to the laboratory in the shortest time possible (approximately 20 min), so the impact of oxygen on the meat samples is negligible. Furthermore, as explained in section 2.1 (collection and preparation of chicken samples), the meat samples were packaged and sealed in PVC bags before being conveyed to the laboratory for analysis. We choose PVC because it is flexible, light, cost-effective, transparent, tough and safe. It has excellent organoleptic properties (does not affect the taste of the packaged food). Kindly find the updated section 2.1 (Collection and preparation of chicken samples) in the revised manuscript for these changes.
Q10: Line 119. Explain where the 80 chicken samples came from. If there are 14 samples x 6 days, there should be 84 samples in total and not 80. Please mention in detail.
Re: Thank you for your valuable and positive suggestion. Actually, during the 11 days in our study, 14 samples were randomly taken out from the refrigerator to determine its TVB-N content every other day (i.e. 1st, 3rd, 5th, 7th, 9th day and 11th day). Therefore, a total of 84 samples were used for the TVB-N analysis. However, the CSA data of four samples was inadvisable due to human error. In addition, in the process of model building, we need to divide the sample into the training and prediction sets according to 3:2. Therefore, we eliminated 4 samples and in total of 80 samples were separated into calibration and prediction sets in a 3:2 ratio. During the analysis of the data sets, 48 samples were allocated to a calibration set for the purpose of constructing the model. The remaining 32 samples were assigned to a prediction set to test the robustness of the model that was constructed. To make reader better understand, as you suggested, we have supplemented more details about the number of samples we used in this study. Kindly find the Lines 130-134 in the revised Mns.
Q11: Line 260. And the values of the TVB-N content during refrigerated storage (1-11 days)?
Re: Thank you for your helpful suggestions to improve this paper. According to your suggestion, we added the values of the TVB-N content during refrigerated storage (1-11 days). Kindly find the Figure S1 and Table S1 in the revised Supporting information.
Q12: Line 263. Why was the total viable count not determined to relate to the volatile nitrogenous compounds produced during refrigerated storage?
Re: We appreciate the thoughtful and helpful comments you made that will help strengthen this text. Indeed, the total viable count (TVC) and total volatile basic nitrogen (TVB-N) are two critical indicators for evaluation the freshness of meat and other aquatic species.
However, in this study, the colorimetric sensor used is constructed based on the response between odor information generated by the meat sample and chemical dyes. Chicken is rich in nutrition and contains a variety of enzymes. The growth and propagation of bacteria and the action of enzymes will cause the odor change of chicken. Therefore, the odor information change of chicken is one of the most important criteria to evaluate its freshness. While in chicken rotten odor, mainly ammonia and amine substances, and these alkaline nitrogen substances collectively known as TVB-N. Therefore, TVB-N is often used as the key index for the freshness evaluation of meat products in national standards. Therefore, this is the main reason why TVB-N was selected to carry out the research in this study. In addition, more importantly, the main principle of the color sensitive sensor constructed in this study to detect the freshness of chicken is based on the response between chicken odor information and dyes. Therefore, the TVB-N that the main nitrogen-containing odors of chicken is selected as the freshness evaluation index, which is directly related to the content of this study.
Certainly, we think that your suggestion is very valuable and very helpful for our future research. In the future, our group is engaged in evaluating the freshness of chicken meat as well as other meats and aquatic life, exploring different indicators such as TBARS, TVC, and TVB-N in conjunction with other approaches. Kindly find the Lines 53-56, 66-72, 104-106 in the revised Mns.
